# Scenario-based forecasting of the global energy demand and carbon footprint of artificial intelligence

Berke M. Turkay[1], Ipek Pehlivan[2], Nuri C. Onat[3], Murat Kucukvar[4], Metin Türkay[2]*

**1** Mitchell E. Daniels, Jr. School of Business, Purdue University, West Lafayette, Indiana, United States of America, **2** Department of Industrial Engineering, Koc University, Istanbul, Turkey, **3** Qatar Transportation and Traffic Safety Center, College of Engineering, Qatar University, Doha, Qatar, **4** Daniels School of Business, University of Denver, Denver, Colorado, United States of America

* mturkay@ku.edu.tr

## Abstract

Artificial intelligence (AI) is advancing rapidly and is emerging as a significant driver of global electricity consumption, yet its long-term energy and emissions implications remain poorly quantified. This study develops a scenario-based, simulation-driven modeling framework that links mathematical representations of AI computational demand with life-cycle carbon accounting for global AI-related energy use and emissions through 2050. We evaluate alternative development pathways that differ in model scale, deployment structure, and electricity mix assumptions. Across all scenarios, improvements in hardware and algorithmic efficiency substantially reduce energy use per operation; however, aggregate AI electricity demand still increases by roughly an order of magnitude due to rapid growth in training and inference workloads. Under the continuation of current trends, AI electricity consumption could reach up to 30% of global demand by 2050, corresponding to more than 8 gigatons of annual $CO_2$-equivalent emissions. Even under optimistic efficiency trajectories, total AI-related electricity demand remains more than six times higher than 2024 levels. In contrast, scenarios that combine consolidation toward fewer, larger models with transitions to low-carbon electricity sources reduce total emissions by up to 40% relative to business-as-usual pathways, exceeding the reductions achievable through efficiency gains alone by more than 20 percentage points. These results highlight widening regional disparities and indicate that policy choices affecting AI deployment patterns and electricity system decarbonization play a central role in shaping the carbon intensity of computation.

## Introduction

AI systems are increasingly integrated into domains such as healthcare, finance, transportation, and scientific research. Despite this expansion, the environmental implications of AI development and deployment remain insufficiently quantified at a

**Data availability statement:** A supplementary information file containing a detailed breakdown of the methodology and sources for raw data for this research is included in the submission.

**Funding:** The author(s) received no specific funding for this work.

**Competing interests:** No authors have competing interests.

global scale. As model architectures grow in parameter count and inference workloads become more pervasive, energy consumption associated with AI systems is projected to rise [4]. Existing studies primarily provide case-specific estimates. For example, training the BERT language model consumed over 1,500 kWh, approximately the annual electricity use of a U.S. household [31], whereas GPT-3's training phase was estimated to emit more than 500 metric tons of $CO_2$-eq, roughly equivalent to the lifetime emissions of five gasoline vehicles [23].

Although illustrative, these figures do not generalize across model types or deployment scales. Data centers, which form the core infrastructure for AI computation, accounted for approximately 4% of U.S. electricity usage in 2016 [28] and about 1% globally in 2020. This global share is projected to increase as AI workloads expand, though estimates vary depending on assumptions regarding efficiency improvements and model scaling trajectories [18]. Recent public statements by AI industry leaders have highlighted potential energy bottlenecks. For instance, OpenAI CEO Sam Altman suggested at the 2024 World Economic Forum that meeting future AI demands may require substantial advances in energy generation [19]. Such remarks underscore the need to move beyond anecdotal claims toward systematic, scenario-based analyses of AI's long-term energy and carbon footprint.

Much of the existing literature has focused on the energy cost of individual training or inference tasks, yet large-scale deployment of trained models increasingly dominates total computational demand. Prior work in sustainable energy systems shows that the climate impact of emerging technologies depends on how efficiency gains interact with broader energy-system pathways, rather than on device-level performance alone [25]. As AI deployment expands across consumer and industrial applications, projections suggest that inference-related energy use may surpass training under high-adoption scenarios by mid-century [34]. Despite continued improvements in hardware and chip efficiency, aggregate compute requirements have grown rapidly, with estimates indicating that the compute required to train frontier models doubled approximately every 3.4 months between 2012 and 2018 [22].

Beyond increasing total electricity demand, AI systems also influence the carbon intensity of computation by shaping when, where, and at what scale electricity is consumed, through choices in model architecture, deployment patterns, and reliance on centralized versus distributed infrastructure. The integration of AI into sectors such as energy systems [16] and circular economy logistics [36] introduces additional pathways for indirect emissions, requiring assessments that extend beyond isolated model-level benchmarks. Geographic variation in electricity generation plays a central role. For example, training the same BERT model in Germany or the Central United States produces more than double the $CO_2$-eq emissions compared to Norway or France, where electricity mixes are less carbon-intensive [2]. Similarly, data centers powered by coal-based grids can emit up to an order of magnitude more GHGs per unit of computation than those supplied by renewables [5]. These disparities motivate the use of spatially resolved carbon accounting frameworks. Multi-regional input-output (MRIO) analysis provides a robust method for incorporating upstream and location-specific emissions into comprehensive estimates of AI's carbon footprint [7,12].

Aggregate energy demand from AI systems is therefore emerging as a non-negligible component of global electricity consumption. Recent projections suggest that the share of data center electricity use could increase to 4.5% by 2030, with U.S. data centers accounting for up to 5% of national electricity consumption by 2028 [20]. Under sustained AI growth, such expansion may intensify competition for energy resources, particularly in regions with capacity-constrained grids. To address these dynamics, this study integrates an MRIO framework with forward-looking mathematical modeling to assess lifecycle energy use and carbon emissions of AI systems. The model links end-use electricity demand, GPU efficiency trends, and upstream hardware manufacturing emissions, enabling dynamic projections of AI-related energy demand under alternative deployment pathways and energy system configurations.

As shown in Fig 3, AI is estimated to have accounted for approximately 0.4% of global electricity consumption in 2024, consistent with previous studies [28]. Under three model-scaling scenarios: (i) *Baseline*, (ii) *More, Smaller Models*, and (iii) *Fewer, Larger Models*, AI's share of electricity use is projected to rise to between 13% and 47% by 2050, depending on demand growth, hardware improvements, and deployment patterns. Each deployment trajectory is evaluated under two electricity mix assumptions: (a) *Business-as-Usual* (BAU), which assumes constant national generation portfolios, and (b) *Energy Target* (ET), which assumes full achievement of stated 2050 renewable goals. The resulting six scenario combinations are summarized in Table 1. By incorporating life-cycle emissions from both energy production and hardware supply chains, the framework enables more disaggregated assessments than approaches limited to operational energy use.

This study addresses a central gap in the literature on AI and environmental sustainability: the absence of an integrated framework that jointly accounts for global AI deployment trajectories, evolving electricity systems, and supply-chain emissions. Prior work typically treats these dimensions in isolation, relying on static benchmarks or localized case studies that cannot capture how AI's environmental footprint emerges across regions and over time. The contribution of this research lies in coupling scenario-based projections of AI computational demand with multi-regional input-output analysis, enabling simultaneous evaluation of operational electricity use and upstream emissions. By explicitly modeling both training and inference phases and linking geographically disaggregated deployment scenarios to national energy mixes, the framework captures spatial and temporal heterogeneity in carbon intensity. Projections across six scenarios further illustrate how AI's share of global electricity demand depends on population growth, technology adoption, and development pathways, providing a basis for evaluating mitigation strategies under alternative decarbonization trajectories.

## Methods

As illustrated in Fig 1, we develop an integrated modeling framework to quantify the environmental impact of AI systems by linking computational demand, energy use, and greenhouse gas (GHG) emissions. The framework decomposes emissions into direct (operational electricity consumption) and indirect (hardware manufacturing and upstream energy production) components. It incorporates three primary modules: (1) a dynamic model of AI-related computational demand,

**Table 1**. Description of AI electricity mix assumptions (rows) and AI deployment scenarios (columns).

| Scenario | Baseline | More smaller models | Fewer larger models |
|---|---|---|---|
| **Business-as-usual (BAU)** | Continuation of current scaling trends with incremental growth and existing energy mixes. | Decentralized deployment with many smaller models, increasing aggregate computational load and operational complexity. | Consolidated deployment with fewer high-capacity models, reducing redundancy but concentrating processing demand regionally. |
| **Energy target (ET)** | Cleaner energy mix combined with current scaling trends and conventional efficiency gains. | Decentralized deployment persists under low-carbon energy, with high system complexity driven by model count. | Centralized deployment paired with low-carbon energy and efficiency improvements, yielding more streamlined resource use. |

Columns represent AI deployment strategies, while rows represent electricity mix assumptions. Each cell summarizes implications for the carbon intensity of computation under the corresponding scenario.

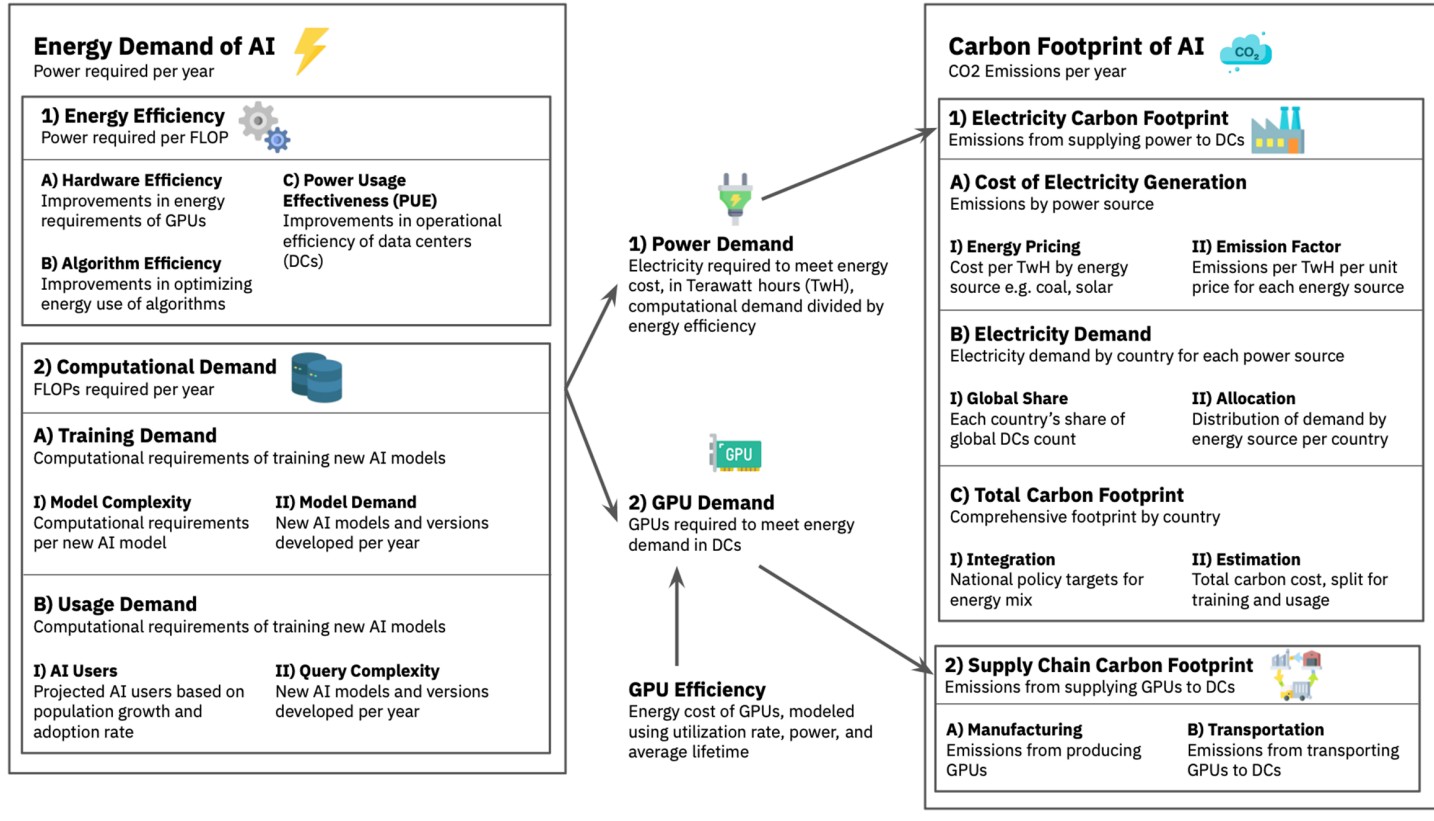

**Fig 1**. **Methodological flow of this work.** Modeling framework for estimating AI's carbon footprint across scenarios.

disaggregated by training and inference; (2) energy efficiency trajectories of GPU hardware; and (3) regional electricity carbon intensities and supply chain dependencies. To estimate total emissions, we combine this structure with a global MRIO model, capturing upstream emissions from hardware production and energy infrastructure. This hybrid approach enables projections of AI's lifecycle carbon footprint under alternative growth and policy scenarios, including variation in model scaling, deployment, and trends in the adoption of renewable energy.

## Energy efficiency

In our model, the total energy consumption of AI infrastructure arises from the relationship between computational demand and energy efficiency. We decompose computational workload into training and inference components: the former scales with model complexity and retraining frequency, whereas the latter is driven by deployment intensity and query volume. Improvements in hardware design, algorithmic processing, and data center infrastructure, including cooling and power conversion systems, determine energy efficiency.

To provide a hardware-agnostic basis for cross-architecture comparison, we adopt watts per floating-point operation (W/FLOP) as the functional unit. This metric enables normalized tracking of energy efficiency trends across heterogeneous AI systems, albeit without capturing rebound effects or upstream material impacts. We define total energy efficiency $E(t)$ as the effective energy required per floating-point operation at time $t$, accounting for three independent dimensions: hardware design ($E_h$), algorithmic optimization ($E_a$), and data center performance as reflected by power usage effectiveness ($E_p$).

We employ exponential functional forms with deceleration to represent efficiency improvements over time. This choice reflects two empirically observed features of technological adoption: rapid early gains driven by innovation and scale, followed by diminishing marginal improvements. Exponential-with-saturation formulations are commonly used to approximate learning curves in hardware performance, algorithmic efficiency, and infrastructure optimization, while preserving analytical tractability over long time horizons. These functional forms are not intended as short-term forecasts, but as stylized representations that bound plausible efficiency trajectories under continued innovation. Parameter values are calibrated using reported historical ranges and industry benchmarks, as documented in Supplementary Sects 2.1–2.3.

**Hardware efficiency ($E_h(t)$).** Hardware efficiency describes the average energy consumed per FLOP by AI-relevant processors, primarily GPUs. We model its improvement using a decelerating exponential form, chosen to reflect empirically observed rapid early gains followed by diminishing returns arising from physical limits such as power density, thermal constraints, and slowing transistor scaling.

$$E_h(t) = E_{h0} \exp\left(\frac{\alpha_h}{\beta_h}\left(1 - \exp(-\beta_h t)\right)\right) \tag{1}$$

- $E_{h0}$: initial efficiency (W/FLOP) at baseline year
- $\alpha_h$: initial rate of efficiency improvement
- $\beta_h$: saturation parameter capturing diminishing marginal gains

Parameter values are calibrated using historical GPU performance-per-watt data spanning multiple architectural generations, as detailed in Supplementary Sect 2.1.

**Algorithmic efficiency ($E_a(t)$.)** Algorithmic efficiency captures reductions in effective computational work per task due to software-level advances, including pruning, quantization, architectural refinements, and solver improvements. As with hardware, algorithmic gains exhibit early rapid progress followed by saturation as problems approach theoretical or practical limits.

$$E_a(t) = E_{a0} \exp\left(\frac{\alpha_a}{\beta_a}\left(1 - \exp(-\beta_a t)\right)\right) \tag{2}$$

- $E_{a0}$: baseline algorithmic energy per FLOP
- $\alpha_a, \beta_a$: parameters governing improvement rate and deceleration

These parameters are estimated from longitudinal solver and algorithm performance benchmarks, described in Supplementary Sect 2.2.

**Power usage effectiveness (PUE, $E_p(t)$).** PUE reflects infrastructure-level efficiency, measuring non-computational overhead from cooling, power conversion, and facility operations. Historical data show substantial early improvements followed by convergence toward physical minima, motivating the same decelerating exponential structure.

$$E_p(t) = E_{p0} \exp\left(\frac{\alpha_p}{\beta_p}\left(1 - \exp(-\beta_p t)\right)\right) \tag{3}$$

- $E_{p0}$: initial PUE at baseline
- $\alpha_p, \beta_p$: infrastructure improvement parameters

Calibration is based on reported industry-wide PUE trends for large-scale data centers, summarized in Supplementary Sect 2.3.

**Aggregate energy efficiency.** The composite energy efficiency $E(t)$, representing net energy per FLOP, is defined as:

$$E(t) = \frac{E_h(t)}{E_p(t) \cdot E_a(t)} \tag{4}$$

This formulation assumes separability between hardware, algorithmic, and infrastructure-level efficiency gains.

## Computational demand

We model total computational demand $C(t)$ as the sum of two components: model training and model usage (inference). This decomposition reflects the distinct temporal, spatial, and operational characteristics of each activity and is standard in systems-level analyses of large-scale AI workloads. Training demand is episodic, highly concentrated, and dominated by the development and retraining of large models, whereas inference demand is continuous, geographically distributed, and driven by deployment scale and user interaction intensity.

$$C(t) = C_t(t) + C_u(t) \tag{5}$$

Training demand $C_t(t)$ is modeled as the product of average model complexity and the number of training runs performed over time. Model complexity captures the total floating-point operations required for a full training cycle, scaling by the number of parameters required to train models. Meanwhile, usage demand $C_u(t)$ is modeled as a function of deployment intensity and query volume. It scales with the number of active users and the average computational cost per query. Unlike training, inference demand exhibits smoother temporal dynamics and is more tightly coupled to adoption patterns, platform integration, and service-level performance requirements. This structure allows inference growth to respond endogenously to user diffusion scenarios without assuming proportional scaling with training activity.

We employ smooth parametric growth functions for both $C_t(t)$ and $C_u(t)$ that reproduce empirically observed scaling behavior while allowing for saturation and deceleration effects. This choice reflects two stylized facts of AI deployment: rapid early expansion driven by capability breakthroughs, followed by slower growth as economic, organizational, and application-level constraints become binding. These functional forms are not intended as short-term forecasts, but as bounded representations of plausible long-run computational demand trajectories under alternative adoption and development scenarios. All parameters governing computational demand are calibrated using reported historical benchmarks on model scale, training frequency, inference workloads, and user adoption, as documented in Supplementary Sects 3.1–3.4.
**Training demand ($C_t(t)$).** Training demand is determined by the product of model complexity and the number of models trained per year:

$$C_t(t) = C_c(t) \cdot C_m(t) \tag{6}$$

Here,

- $C_c(t)$: average model complexity, measured in floating-point operations (FLOPs) per training cycle
- $C_m(t)$: number of models trained annually

Model complexity is represented using a power-law approximation calibrated to historical benchmarks of large-scale model training. Power-law scaling captures sustained growth with sub-exponential curvature, consistent with observed increases in parameter counts and training compute over the past decade:

$$C_c(t) = C_{c0} \cdot (t + \lambda_{c1})^{\eta_{c1}} \tag{7}$$

Similarly, model proliferation is represented as:

$$C_m(t) = C_{m0} \cdot (t + \lambda_{m1})^{\eta_{m1}} \tag{8}$$

This formulation allows training demand to grow superlinearly over time while avoiding unconstrained exponential divergence. All parameters ($C_{c0}, C_{m0}, \lambda_{c1}, \lambda_{m1}, \eta_{c1}, \eta_{m1}$) are empirically fitted using published training benchmarks and industry-reported model counts. Data sources, calibration ranges, and sensitivity bounds are documented in Supplementary Sect 3.2, with implementation provided in the `ai-training` branch of the repository.

**Usage demand ($C_u(t)$).** Usage demand arises from the execution of trained models and is scaled across global users, adoption dynamics, and query-level computational intensity:

$$C_u(t) = \sum_{i=1}^{n} \left(P_{i0} e^{r_i t}\right) \cdot A_i(t) \cdot \left(Q_0 e^{\chi t}\right) \tag{9}$$

- $P_{i0}$: baseline population of user group $i$
- $r_i$: annualized population growth rate in region $i$
- $A_i(t)$: adoption rate of AI services, modeled as:

$$A_i(t) = \frac{1}{1 + \exp\left(-\gamma_i(t - t_{0i})\right)} \tag{10}$$

- $Q_0$: baseline query complexity (FLOPs/query)
- $\chi$: growth rate of query-level computational intensity

Population growth is modeled exponentially to reflect standard demographic projections, while adoption follows logistic diffusion to capture saturation effects and heterogeneous uptake across regions. Growth in query complexity reflects increasing model size, longer context windows, and richer inference tasks rather than assuming uniform expansion in query volume alone. Together, these components produce a flexible representation of usage-driven demand that can be stress-tested across alternative growth assumptions. Supplementary Sect 3.3 provides detailed data inputs, regional segmentation, and uncertainty bounds. Code implementation is available in the `ai-usage` branch.

**Total power consumption.** Given computational demand $C(t)$ and net energy efficiency $E(t)$, total power consumption by AI systems is given as:

$$P(t) = \frac{C(t)}{E(t)} \tag{11}$$

This function is evaluated annually to produce time-indexed electricity demand profiles. For context, projected global power generation is estimated via an exponential regression fitted to International Energy Agency (IEA) data from 1990-2024 (see Supplementary Sect 4.1). Simulations are performed using the primary model file located in the `main-simulation.py` script.

## Estimating carbon footprint

Following the estimation of AI's energy demand, this section quantifies the resulting carbon dioxide equivalent ($CO_2$-eq) emissions. These emissions originate from three identifiable sources: (1) the manufacturing of AI-specific hardware, (2) the international transportation of semiconductor components, and (3) the electricity consumed during model training and inference. Each source is treated as analytically distinct to avoid conflation of upstream (embodied) and downstream (operational) impacts. The methodology for estimating emissions from each category is outlined below.

**Carbon emissions from electricity use.** To quantify emissions from electricity use, projected AI energy demand is geographically distributed based on national shares of global data center capacity. Specifically, we focus on 15 countries that currently account for 82.7% of the world's data center infrastructure [6]. Within each country, energy demand is allocated between training and inference based on modeled workload projections.

Country-level electricity demand is disaggregated into renewable and non-renewable sources using official national energy plans and stated targets for 2030 and 2050. This breakdown maintains current fuel mix proportions unless explicitly projected to change by the respective national authority. This enables consistency across temporal comparisons and corresponds with declared policy trajectories, although the model does not presuppose their fulfillment.

**Multi-regional input-output analysis.** To estimate $CO_2$-eq emissions per unit of electricity generated, we employ a multi-regional input-output (MRIO) model, which explicitly represents inter-industry and cross-border economic linkages across regions [35]. The MRIO framework captures not only the direct emissions arising from electricity generation technologies (e.g., fuel combustion at power plants), but also the indirect emissions embedded along upstream supply chains, including fuel extraction and processing, equipment manufacturing, infrastructure development, and international trade in intermediate inputs. By tracing monetary flows between sectors and regions and linking them with sector-specific emission intensities, the MRIO model enables a consumption-based accounting of electricity-related emissions that avoids truncation errors common in process-based approaches. This comprehensive representation of global production networks is particularly important for electricity systems that rely on imported fuels, technologies, or capital goods, as it allows the attribution of emissions occurring outside national borders to domestic electricity consumption. As a result, the MRIO approach provides a consistent and system-wide estimate of $CO_2$-eq emissions per unit of electricity that reflects the full life-cycle and international dimensions of electricity production [9,15].

Emission factors are then applied to each fuel type. These values, derived from peer-reviewed life-cycle assessments, reflect the full global warming potential (GWP) of energy sources, accounting for combustion emissions, infrastructure development, upstream extraction, and distribution. Emission factors are not uniform across countries, as energy mix compositions and infrastructure efficiency vary. Import dependencies are also incorporated, ensuring emissions from externally sourced electricity are attributed to the importing country. The final output is a matrix of country and fuel-specific emission intensities, which is used to compute the total $CO_2$-eq emissions associated with AI's projected electricity use under multiple scenarios.

The life-cycle greenhouse gas (GHG) emissions attributable to electricity generation and GPU manufacturing are estimated using a global MRIO model. This analysis utilizes the EXIOBASE 3.8.2 database [30], which contains environmental and economic data across multiple countries and sectors. Electricity-related emissions are computed based on national electricity consumption associated with data centers, whereas emissions from GPU manufacturing are attributed to production facilities located in Taiwan, reflecting prevailing global semiconductor supply patterns. The sectoral classifications employed in the emission calculations are provided in S10–S15 Tables of the Supporting Information (SI). Key data inputs for the MRIO framework, including electricity production values, emission factors, price tables, and country-specific adjustments, are summarized in this section, while detailed numerical tables and sectoral mappings are provided in the Supplementary Information file.

The MRIO model is formulated according to Leontief's input-output framework. Total gross output $x$ is calculated as:

$$x = (I - A)^{-1} y \tag{12}$$

where $x$ is the gross output vector (in million euros), $I$ is the identity matrix, $A$ is the matrix of direct input coefficients, and $y$ is the final demand vector. The term $(I-A)^{-1}$, known as the Leontief inverse and denoted $L$, represents the total input requirements (direct and indirect) per unit of final demand. To quantify environmental burdens, the MRIO framework incorporates an environmental satellite account matrix $E$, which contains sector-specific GHG emissions per unit of

economic output. The emission intensity matrix $B$ is then defined as:

$$B = E \cdot (\mathrm{diag}(x))^{-1} \tag{13}$$

where $B$ expresses the GHG intensity in kg $CO_2$-eq per monetary unit of sectoral output, and $x$ denotes the economic output vector for each sector.

To determine emissions attributable to a specific final demand vector $y$, the model computes:

$$e_f = BLy \tag{14}$$

where $e_f$ is the resulting emissions vector (in kg $CO_2$-eq/M€), capturing both direct emissions and indirect upstream supply chain impacts associated with the production and delivery of goods and services. This analysis enables a thorough estimate of AI-related emissions from both operational electricity use and upstream hardware manufacturing.

**Manufacturing-related carbon footprint.** The manufacturing of GPUs contributes considerably to the life-cycle carbon footprint of AI systems. This analysis attributes GPU production emissions to Taiwan, which currently dominates global advanced semiconductor fabrication. Although potential geopolitical or supply chain shifts could alter this concentration, Taiwan's continued primacy is assumed for modeling consistency, based on its entrenched technological infrastructure and leading-edge lithography capabilities.

For tractability, we assume that average chip sizes will remain approximately constant over the modeled period. This assumption is grounded in known physical constraints on photolithography and transistor miniaturization, which limit further shrinkage in node sizes [17]. Accordingly, per-unit manufacturing emissions are treated as time-invariant.

Manufacturing-related emissions are calculated using an emission factor $e_f$ obtained from the MRIO framework:

$$\text{Manufacturing Emissions} = \text{GPU Demand} \times \text{Cost per GPU} \times e_f \tag{15}$$

where:

- **GPU Demand** represents the total number of units required under a given energy demand scenario.
- **Cost per GPU** is the monetary manufacturing cost of a single unit, based on market and industry reports.
- $e_f$ denotes the emission factor (in kg $CO_2$-eq/€), derived from the MRIO model based on the economic output and carbon intensity of Taiwan's semiconductor sector.

The number of GPUs required is estimated by:

$$\text{GPU Demand} = \frac{P(t)}{\text{GPU Power} \times \text{Utilization Rate} \times \text{GPU Lifetime}} \tag{16}$$

where:

- **GPU Power** is the rated power consumption per unit (700 W) [21]
- **Utilization Rate** reflects average operational intensity across deployments (assumed at 80%) [10]
- **GPU Lifetime** is set at two years, corresponding to typical replacement cycles in data centers [28]

**Transportation-related carbon footprint.** The distribution of AI hardware contributes to global carbon emissions through long-distance logistics, particularly via air freight. Given the geographic concentration of GPU manufacturing in Taiwan

and the global dispersion of data centers, most units must be transported across intercontinental distances. This analysis assumes that air transport remains the dominant mode for GPU delivery due to its speed and logistical reliability, especially for high-value components, and because electrified alternatives for long-haul aviation are not yet commercially viable at scale.

We compute transportation-related emissions as a function of the number of GPU units shipped, the average transport distance per unit, and the modal emission intensity per ton-kilometer. Emission factors incorporate both direct combustion emissions and upstream fuel-cycle effects. Country-specific adjustments are applied based on regional air cargo routes and import dependency. Maritime shipping is excluded due to limited usage in GPU distribution, which prioritizes minimal transit time and high-value safeguarding.

**Training and usage-related carbon footprint.** This section quantifies the carbon emissions arising from AI model training and usage by integrating fuel-specific energy demand with corresponding emission factors across multiple scenarios. The methodology follows a transparent and modular structure comprising the following steps. First, scenario-specific emissions for each fuel type $f$, in each country $c$, under each scenario $s$, are calculated by multiplying the energy demand $D_{f,c,s}$ by the emission factor $e_{f,f}$, which denotes the life-cycle $CO_2$-eq emissions per unit energy from fuel type $f$:

$$E_{f,c,s} = D_{f,c,s} \times e_{f,f} \tag{17}$$

Next, aggregate emissions at the national level are obtained by summing over all fuel types:

$$E_{c,s} = \sum_f E_{f,c,s} \tag{18}$$

Total emissions for each country are then partitioned into training-related and usage-related components using scenario-specific attribution factors $P_{\text{usage},s}$, which represent the share of operational (inference) energy consumption. Training emissions are modeled as the complementary share:

$$E_{c,s}^{\text{usage}} = E_{c,s} \times P_{\text{usage},s} \tag{19}$$

$$E_{c,s}^{\text{training}} = E_{c,s} \times (1 - P_{\text{usage},s}) \tag{20}$$

Finally, total emissions across all countries are obtained by summing country-level values:

$$E_s^{\text{usage}} = \sum_c E_{c,s}^{\text{usage}}, \qquad E_s^{\text{training}} = \sum_c E_{c,s}^{\text{training}} \tag{21}$$

Here, $E_s^{\text{usage}}$ and $E_s^{\text{training}}$ represent the total operational and training-related emissions, respectively, under scenario $s$. The results for each country and scenario (Baseline, Fewer-Larger, and More-Smaller) are documented in the Supporting information (Sects 7.2.4–8.7).

## Total carbon footprint by scenario

This section estimates total carbon emissions by summing contributions from manufacturing, transportation, model training, and operational usage. Emissions are first calculated by multiplying fuel-specific energy demand with corresponding emission factors, yielding country-level estimates of $CO_2$-eq emissions across scenarios. The methodology proceeds as follows: (1) compute fuel-type emissions using previously defined GHG emission factors, (2) aggregate emissions across all fuels within each country, and (3) disaggregate total emissions into components associated with model training and

inference. This accounting framework enables a scenario-sensitive quantification of AI's projected carbon footprint over time. Complete numerical results are provided in Supplementary Sects 9–10.

To evaluate how policy design influences AI's long-term carbon footprint, we analyze three AI deployment configurations under two policy environments: Business-as-Usual (BAU) and Energy Targets (ET). The baseline scenario reflects standard growth in AI model size and usage, assuming no structural change in system design. The *Fewer-Larger* strategy involves the consolidation of workloads into a smaller number of high-capacity models, reducing redundant computations. In contrast, the *More-Smaller* scenario emphasizes flexibility by virtue of increased deployment of smaller models, leading to higher cumulative energy demand. Each deployment path is analyzed under BAU conditions, assuming current energy trends continue, and under ET policies, which model reductions in carbon intensity due to targeted energy transition efforts. These combined evaluations illustrate how both technical configuration and policy orientation affect aggregate emissions.

## Results

### Global trends in energy demand of AI: Potential future scenarios

This analysis projects that, under the assumption of continued current growth trajectories, AI-related emissions could increase by approximately 16.6-fold by 2050 relative to 2030 levels. This rise is attributable to increased computational demand that may outpace improvements in hardware efficiency or reductions in carbon intensity. Fig 2 presents the results from a multi-scenario modeling framework that combines forecasts of AI model scale, algorithmic complexity, and energy consumption under varying deployment strategies.

Computational demand from AI systems is projected to escalate sharply by 2050, with growth driven not only by model proliferation but by the increasing scale of individual architectures. The *Fewer-Larger Models* scenario yields steeper aggregate demand due to concentrated training workloads, whereas the *More-Smaller Models* configuration distributes computational load across a larger number of lighter-weight deployments. Despite lower per-model complexity, the latter results in greater total FLOPs due to frequency and redundancy effects. This divergence extends to projected electricity consumption, as centralized scaling produces localized energy hotspots, whereas decentralized diffusion generates a broader, more distributed power footprint. This indicates that model deployment, not just hardware and algorithm efficiency, will also shape AI's long-run energy demand (Fig 2a–2d).

The projected share of AI in global electricity consumption rises sharply under all adoption scenarios, indicating the sector's potential to become a dominant driver of global energy demand. By 2050, AI workloads are estimated to account for 23% of global electricity consumption under the baseline trajectory, a figure that rivals historically energy-intensive sectors. The *More-Smaller Models* scenario, although premised on individually efficient deployments, amplifies aggregate demand via sheer volume, pushing AI's electricity share above 42% under high-adoption conditions. In contrast, the *Fewer-Larger Models* configuration, by concentrating workloads into fewer systems, contains electricity demand to roughly 11%, revealing that deployment architecture imposes a structural ceiling on sectoral energy intensity (Fig 3).

These scenario-dependent outcomes also reflect regional disparities in emissions intensity, as electricity generation profiles vary substantially across national contexts. Countries with carbon-intensive grids, dominated by coal or natural gas, will produce higher marginal emissions from incremental increases in AI demand. In contrast, regions with high renewable penetration may experience lower associated emissions, assuming renewable capacity scales proportionally with increased demand. These dynamics suggest that decarbonization trajectories and grid resilience will play critical roles in shaping the net environmental impact of AI expansion.

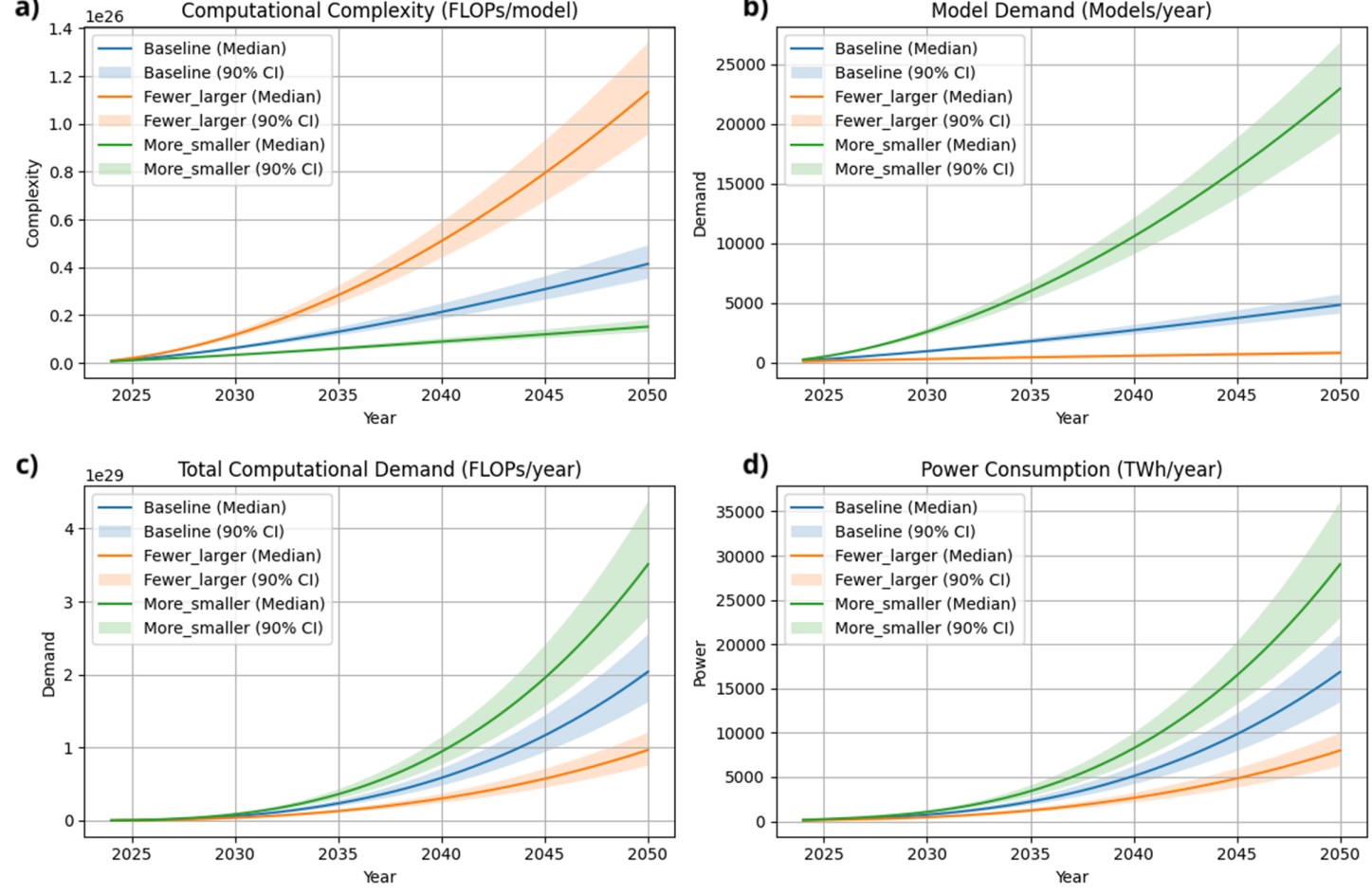

**Fig 2**. **Illustration of projected trends in AI computational and energy demand under different scaling scenarios from 2024 to 2050.** Projected AI model growth and energy demand under three scaling scenarios (2024-2050). Panel (a): computational complexity per model (FLOPs). Panel (b): annual model deployments. Panel (c): total computational demand. Panel (d): projected power consumption (TWh/year) for training and inference. Curves show median projections; shaded bands denote 90% confidence intervals. Scenarios include baseline, fewer larger models, and more smaller models.

## Carbon footprint of AI: Life-cycle breakdown and future scenarios

This analysis evaluates the carbon footprint of AI across four distinct phases: hardware manufacturing, international transportation, model training, and operational usage. Under the baseline scenario, manufacturing emissions are projected to increase from 0.029 Gt $CO_2$-eq in 2024 to 2.79 Gt by 2050. Scenario variation introduces noteworthy divergence: the *Fewer Larger Models* case reduces cumulative manufacturing emissions to 1.63 Gt, whereas the *More Smaller Models* trajectory results in a projected 4.28 Gt. Transportation-related emissions indicate comparable scaling, rising from 0.0267 Gt in the baseline to 0.13 Gt under the higher-distribution scenario. Training emissions grow from 0.30 Gt in 2030 to 4.29 Gt in 2050, whereas usage-phase emissions rise from 0.07 to 1.21 Gt within the same period. These projections suggest that emission outcomes are sensitive to scaling configurations and indicate the relevance of model architecture choices and energy system decarbonization for long-term climate impacts.

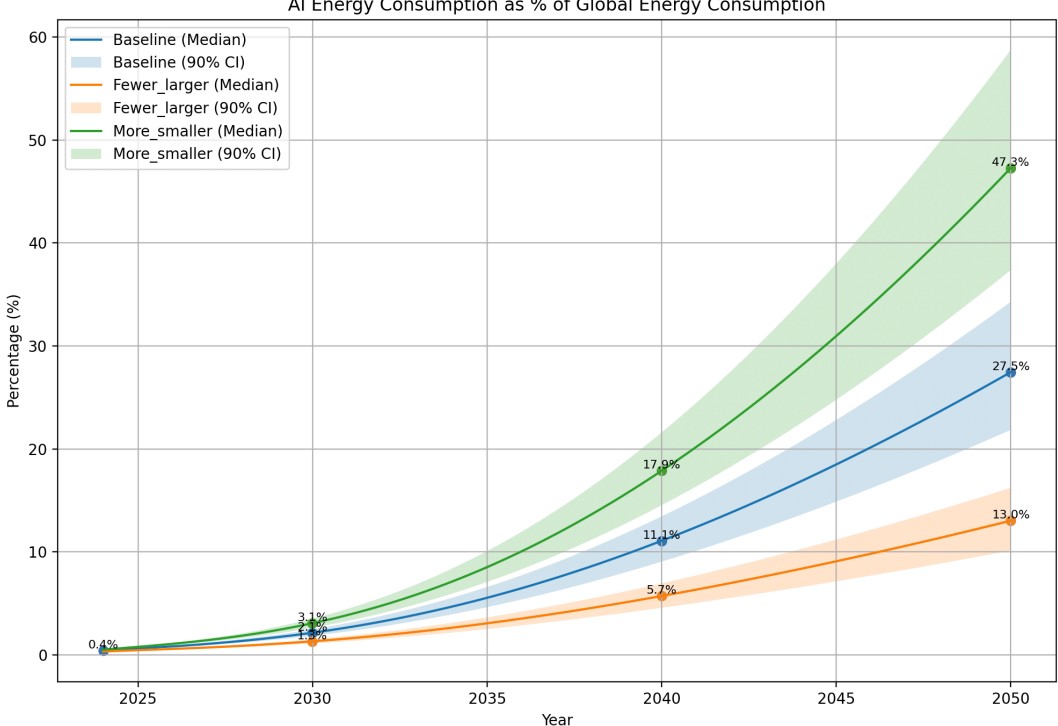

**Fig 3**. **Projected trajectory of AI's share of global electricity consumption under different scaling scenarios (2024-2050).** Projected share of global electricity demand from AI (2024-2050) under baseline, fewer larger models, and more smaller models scenarios. Curves show median forecasts; shaded bands indicate 90% confidence intervals.

## Carbon footprint comparison by scenario

Scenario-based projections reveal substantial divergence in AI-related emissions by 2050, with outcomes highly sensitive to both deployment architecture and energy policy trajectory. Under the *Business-as-Usual* scenario, the *More-Smaller Models* configuration produces the highest carbon burden, reaching 22.05 Gt $CO_2$-eq, whereas the adoption of *Energy Target* policies moderates this total to 12.84 Gt, a 41.8% reduction. By contrast, the *Fewer-Larger Models* pathway yields the lowest emissions across both policy regimes, at 6.61 Gt under BAU and 4.86 Gt under ET. Across all scenarios, training dominates the emissions profile, comprising up to 68.5% of total output under BAU. These results crystallize the structural trade-offs inherent in AI system design: widespread model diffusion amplifies cumulative demand despite per-unit efficiency, while consolidation constrains emissions but may limit flexibility and domain-specific performance (Fig 4a–4b).

The magnitude and growth trajectories of AI-related electricity demand and emissions projected in this study are consistent with recent sectoral assessments of data centers and digital infrastructure. Under Business-as-Usual conditions, total AI-related emissions reach the order of tens of gigatons of $CO_2$-eq by mid-century, with training activities accounting for the dominant share of emissions across scenarios. Adoption of Energy Target pathways substantially reduces total emissions, particularly in scenarios that consolidate training workloads. Detailed country-level and phase-resolved results underlying these patterns are reported in the Supporting information (S35–S44 Tables).

## Global distribution of AI energy demand

To assess variation in projected AI adoption, countries are stratified into five tiers based on their Inequality-Adjusted Human Development Index (IHDI). The choice of IHDI, rather than GDP per capita or aggregate digital investment,

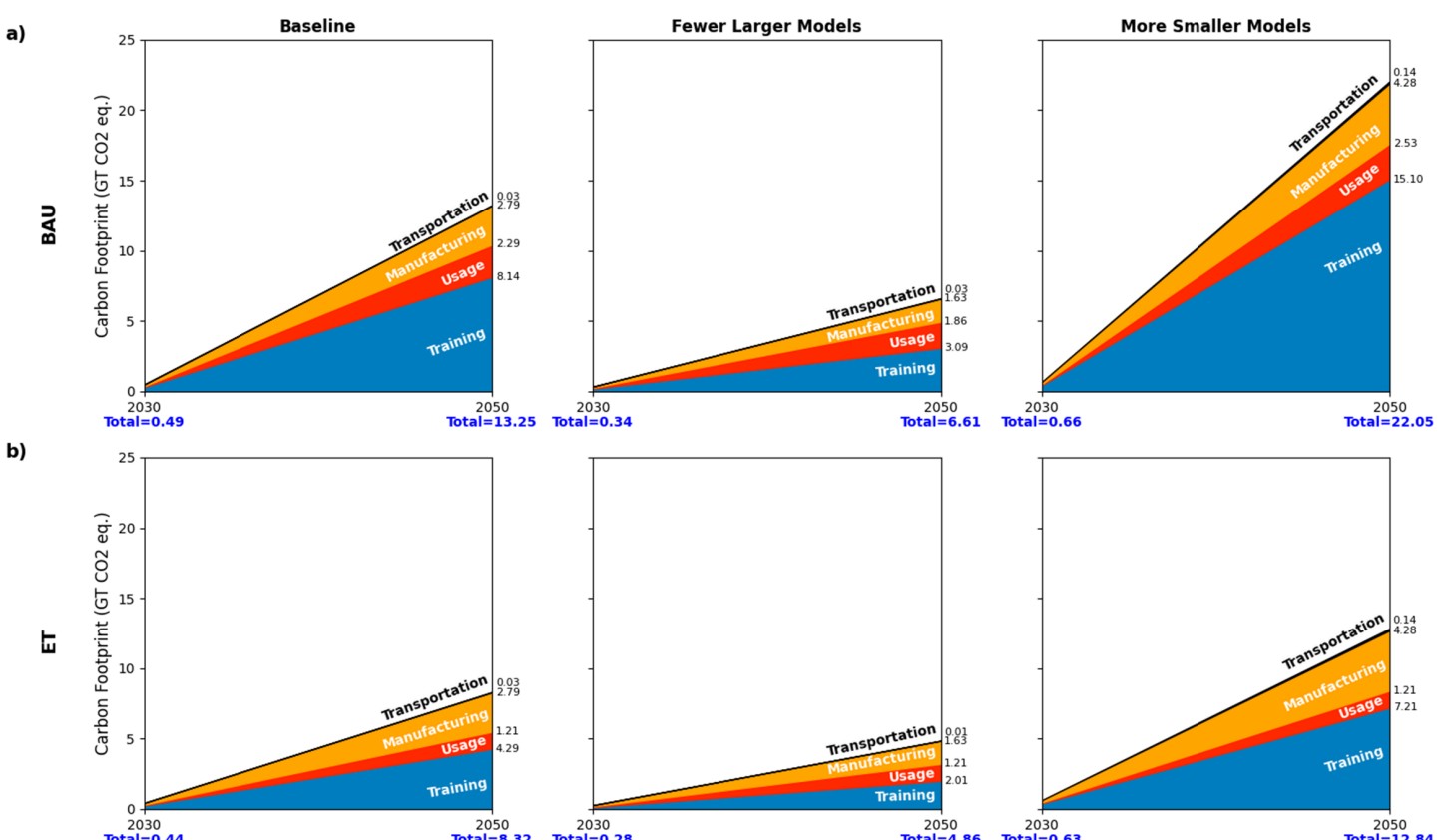

**Fig 4**. **Projected AI carbon footprint by electricity mix and AI energy demand scenarios for 2030 and 2050.** Projected AI carbon footprint in 2030 and 2050 under different electricity mix and model scaling scenarios. Panel (a) presents Business-as-usual (BAU) outcomes; panel (b) shows Energy Target (ET) projections. Each includes three deployment strategies: baseline, fewer larger models, and more smaller models. Emissions are disaggregated by lifecycle phase: training, usage, manufacturing, and transportation.

reflects a methodological preference for a composite indicator that incorporates both average development and internal inequality. Unlike raw income-based metrics, the IHDI adjusts for disparities in education, health, and income distribution, offering a more holistic proxy for national capacity to absorb and integrate AI technologies. This adjustment is particularly important for capturing the extent to which development is equitably distributed and structurally embedded, key determinants of sustained digital infrastructure growth and skilled labor availability.

Projected adoption trajectories reveal a widening stratification in global AI integration, with national capacity tightly coupled to developmental asymmetries. Countries in the highest IHDI quintile are expected to exceed 80% AI adoption by the mid-2030s, reaching near-saturation by 2050. These states possess structural advantages in broadband infrastructure, AI research ecosystems, and access to high-performance computing, enabling more rapid adoption [34]. In contrast, countries in the lowest IHDI tier remain locked into trajectories of marginal adoption, projected to stay below 30% by 2050. Persistent deficits in digital capital, technical education, and infrastructure resilience constrain both access and absorptive capacity, limiting the developmental reach of AI and reinforcing the uneven geography of digital modernization [34].

The divergence in adoption patterns reinforces the persistence of structural asymmetries in the global distribution of digital capability. These disparities parallel earlier technological gaps observed in the evolution of the global

digital divide [11], raising concerns that the benefits of AI may similarly accrue along existing lines of socioeconomic stratification.

### Geographies of disproportion: national contributions to AI's carbon footprint

National disparities in AI-related emissions are shaped not merely by user counts but by the geographic concentration of computation and the carbon intensity of local energy systems. Countries with advanced infrastructure and large-scale data center deployment disproportionately bear the emissions burden, even when their share of end users is relatively modest. Emissions are modeled as a function of projected electricity demand and the carbon intensity of domestic power generation, adjusted for the spatial distribution of compute workloads. Adoption rates are inferred using structural proxies such as broadband penetration, per capita GDP, and AI investment levels, whereas computational demand per user reflects both deployment architecture and usage intensity. The result is a global asymmetry in which energy infrastructure has a greater impact on national contributions to AI's carbon footprint than population or adoption rate (Fig 6).

The United States is projected to display a disproportionate emissions profile. By 2030, it is estimated to account for 11.7% of global AI users, above its global population share, despite contributing approximately 66% of AI-related emissions. This discrepancy arises primarily from two quantifiable drivers. First, the U.S. hosts an estimated 45% of global data center capacity, meaning that a substantial volume of international AI workloads is processed domestically, concentrating energy use [28]. Second, U.S. decarbonization pathways lag relative to peers, with projections indicating sustained reliance on fossil-fuel-based electricity, particularly coal and natural gas, by 2050 [13,29].

Germany illustrates a comparatively lower-emissions trajectory. In 2030, it is projected to represent 2.9% of global AI users despite contributing only 1.9% of modeled emissions. This efficiency is partly attributable to Germany's continued investment in renewable energy and policy commitment to a net-zero electricity system by the mid-2040s [1]. However, cross-border data routing introduces complexity in attribution: a portion of German AI queries are processed in foreign infrastructure, particularly U.S. facilities, potentially externalizing emissions. These results emphasize that the carbon footprint of AI services is shaped not only by national adoption rates but also by infrastructure geography and decarbonization trajectories.

## Discussion

### Beyond efficiency: Policy, innovation, and structural interventions for sustainable AI

Technological innovation continues to improve the energy efficiency of AI systems. Algorithmic techniques such as sparse modeling and knowledge distillation have demonstrated measurable reductions in computational load without substantial performance degradation [33]. Concurrently, hardware developments, including domain-specific accelerators and neuromorphic architectures, improve energy-per-operation efficiency [14]. At the infrastructure level, data center designs incorporating waste heat recovery, direct liquid cooling, and high renewable-energy penetration can further reduce the energy intensity of AI service provision [28]. These trends are reflected in the declining energy-per-FLOP trajectories observed across all scenarios (see Supplementary S1–S3 Figs).

However, scenario-based modeling indicates that efficiency improvements alone are insufficient to offset projected growth in AI adoption and model complexity. Under the baseline trajectory, total AI-related emissions increase by a factor of 16.6 by 2050 relative to 2024 levels (Fig 4a). This outcome arises because exponential growth in computational demand consistently outpaces realized and anticipated efficiency gains across all modeled deployment pathways (see Supplementary S4–S11 Figs), consistent with prior assessments of rebound effects in energy-intensive digital systems [18]. In contrast, scenarios that combine efficiency improvements with structural changes, including low-carbon electricity deployment and alternative model scaling strategies, achieve materially larger emissions reductions (Fig 4b).

Mitigation at scale therefore requires interventions beyond algorithmic or hardware optimization, including electricity system decarbonization, regulation of AI infrastructure siting and operations, and market mechanisms that internalize carbon costs [3,13].

Addressing the environmental implications of AI thus necessitates an integrated policy and technology framework aligned with the temporal and structural drivers identified in the model. Governments can mandate emissions disclosure across the AI development pipeline and introduce minimum efficiency standards to reduce near-term impacts [3,18]. Financial instruments, such as tax incentives for renewable-powered compute or penalties on energy-intensive practices, can shift deployment toward lower-impact configurations [27], as illustrated by the divergence between Business-as-Usual and Energy Target scenarios (Fig 4). Industry commitments to carbon neutrality, whether via direct procurement of clean energy or verified offsets, provide additional mitigation channels, though their effectiveness depends on alignment with long-run deployment trajectories rather than efficiency gains alone [3,23].

### Embodied emissions in the AI supply chain: GPU manufacturing and transportation

The environmental burden of AI extends beyond operational emissions from training and inference to include upstream emissions embedded in the production and distribution of specialized hardware. Prior life-cycle assessments identify semiconductor and GPU manufacturing as dominant contributors to the total carbon footprint of ICT systems [26], whereas transportation contributes a smaller but non-negligible share. Consistent with these findings, our results show that manufacturing emissions dominate transportation emissions across all scenarios and time horizons, accounting for the majority of non-operational emissions in both 2030 and 2050 projections (Fig 4). Increased demand for semiconductors, driven by AI-related consumer products and data center expansion, leads to sharply rising manufacturing emissions, particularly in regions with high grid carbon intensity such as China and Indonesia [8].

Under the baseline deployment trajectory, GPU manufacturing emissions increase by more than an order of magnitude between 2024 and 2050, whereas transportation-related emissions remain comparatively small across scenarios (see Supplementary S6–S7 Tables). Differences in AI deployment structure materially affect embodied emissions: scenarios emphasizing fewer, larger models substantially reduce manufacturing-related emissions relative to decentralized deployment with many smaller models (Fig 4), indicating that upstream impacts are sensitive to architectural and scaling choices rather than demand growth alone.

This analysis excludes emissions from end-of-life disposal due to limited data availability and high uncertainty surrounding global hardware reuse, recycling rates, and secondary markets. Nevertheless, strategies aligned with circular economy principles, including modular design, e-waste recovery, and disassembly-oriented engineering, represent potential long-term mitigation pathways not quantified in this study [8]. Given that manufacturing emissions exceed transportation emissions by roughly two orders of magnitude in our projections (Supplementary S6–S7 Tables), circular economy interventions would need to primarily target fabrication-stage processes to achieve material reductions. When combined with AI-enabled design optimization and advancements in low-carbon material science, such interventions could reduce embodied emissions in major production hubs, including Taiwan (see Supplementary S45 Table).

Policy, technological innovation, and supply chain restructuring must therefore be coordinated to achieve meaningful decarbonization of AI hardware. Geopolitical shifts in semiconductor manufacturing are already underway: for example, the Taiwan Semiconductor Manufacturing Company (TSMC) is constructing a fabrication facility in Arizona, while Samsung is expanding operations in Texas [24,32]. These developments may modestly reduce transport-related emissions associated with trans-Pacific logistics, given the geographic co-location of most data centers in North America, but their larger significance lies in the potential to alter the carbon intensity of manufacturing through changes in electricity sourcing and regulatory environments.

### The "global AI divide": Diverging patterns in access and emissions attribution

We use the term "global AI divide" to describe emerging asymmetries in access to AI technologies and in the geographic distribution of their associated carbon burdens. While these patterns parallel earlier forms of digital inequality, they introduce additional dimensions linked to infrastructural dependencies and the spatial separation between AI usage and computation. Accordingly, claims about inclusion and equity must be evaluated against empirical evidence on AI deployment, energy use, and emissions attribution rather than normative expectations alone.

Our modeling results indicate that traditional asymmetries between production and consumption do not fully carry over to the AI domain. As shown in Fig 5, AI adoption is projected to expand rapidly in lower- and middle-income country groups, narrowing gaps in user access over time. At the same time, emissions remain unevenly distributed across countries hosting energy-intensive data center infrastructure. For example, China is projected to account for 41.3% of global AI users by 2030, yet only 6.6% of associated AI-related emissions, while India and Indonesia collectively represent 25.4% of projected AI users but contribute a substantially smaller share of global emissions (Fig 6a). These disparities arise from the geographic concentration of large-scale training workloads and differences in electricity carbon intensity, rather than from AI usage patterns alone.

Looking ahead, continued investment in domestic AI infrastructure may shift some emerging economies from peripheral consumers toward more central roles within the global AI ecosystem. China and India are already expanding local

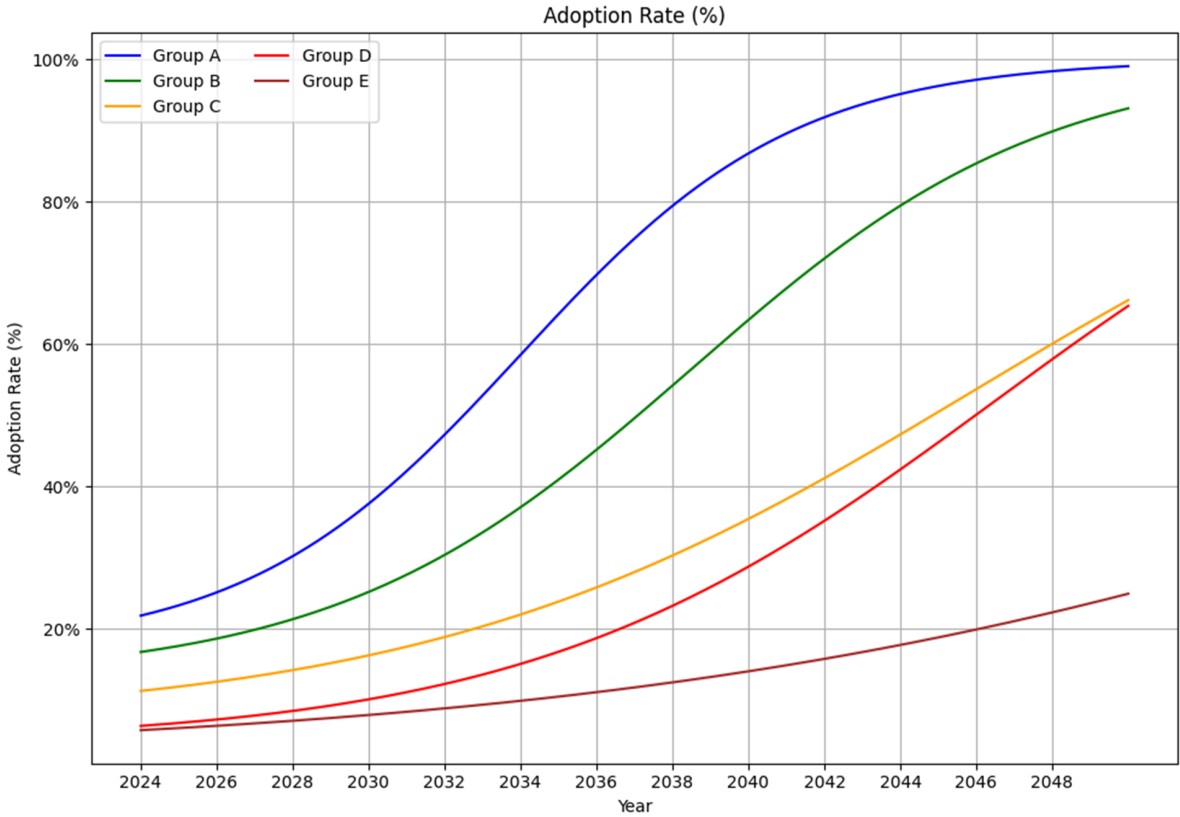

**Fig 5**. **Projected AI adoption rates by country group, stratified by global inequality-adjusted human development index (IHDI), 2024-2050.** Projected AI adoption rates (2024-2050) across five country groups stratified by inequality-adjusted human development index (IHDI). Group A includes the top 20% of countries by human development, while Groups B-E represent progressively lower levels. Adoption follows logistic growth, with faster uptake in higher-IHDI countries.

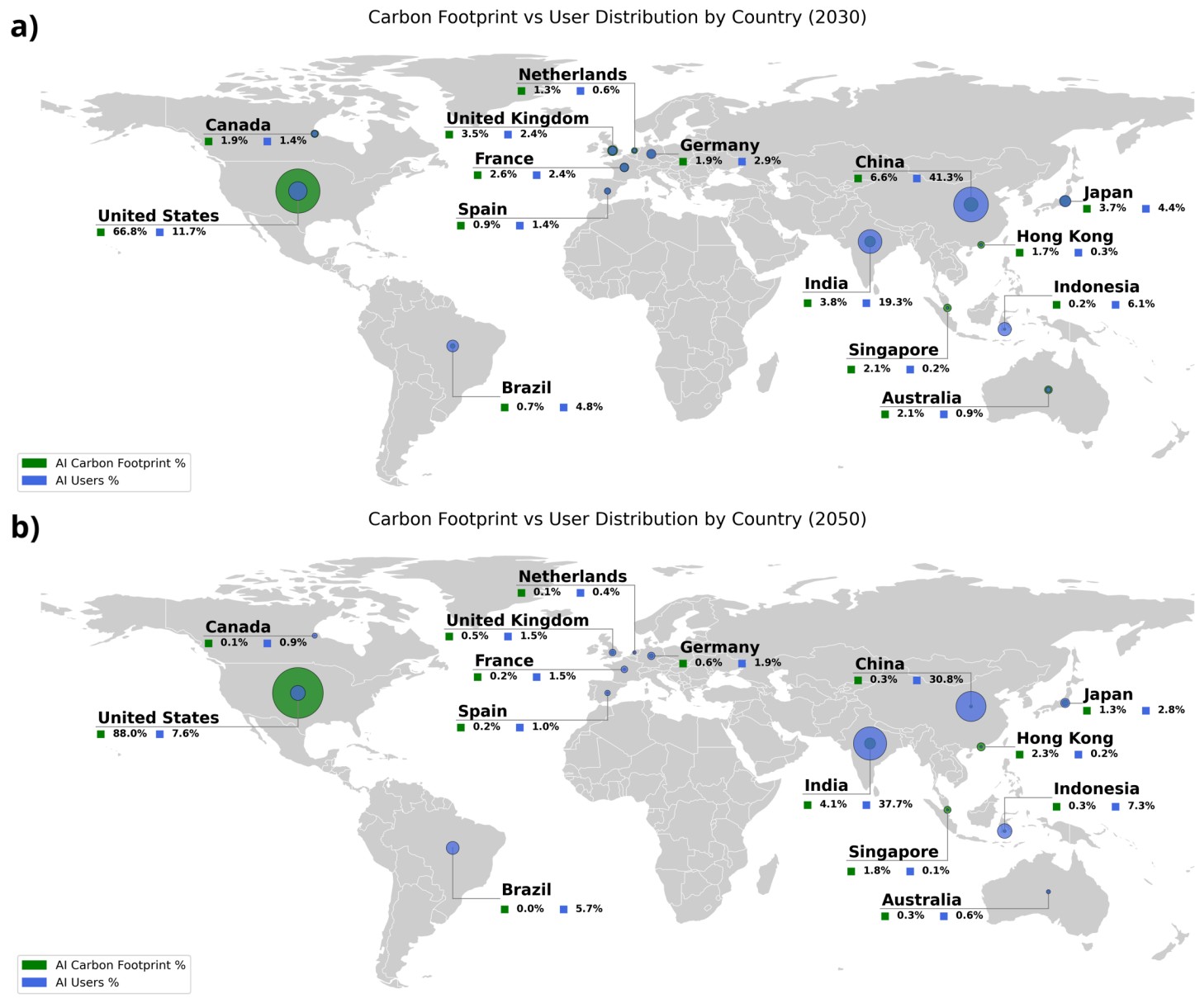

**Fig 6**. **Global distribution of AI user shares and corresponding carbon burden across selected countries for 2030 and 2050.** Panels (a) and (b) depict each country's share of global carbon footprint of AI (green) among the countries analyzed for 2030 and 2050 respectively. In all panels, each country's share of global AI users is shown in blue, with the size of each bubble corresponding to a larger share.

compute capacity, while economies such as Singapore and Hong Kong exhibit high per-capita AI engagement due to their roles as regional digital and financial hubs (Fig 6b). Whether these trends lead to more equitable emissions responsibility depends on the pace of infrastructure localization, regulatory coordination, and grid decarbonization across regions. Given the transnational nature of AI service provision and the divergence in national energy systems, attributing emissions in global AI workflows remains a nontrivial challenge that will require harmonized reporting standards and life-cycle accounting methodologies.

## Conclusion

This study develops a scenario-based, simulation-driven framework to quantify the long-term energy use and carbon footprint of artificial intelligence across regions and deployment pathways. By coupling projections of AI computational demand with electricity system trajectories and multi-regional input–output analysis, the results demonstrate that AI's environmental impact is not determined by efficiency improvements alone, but by the interaction between deployment structure, electricity decarbonization, and the geographic concentration of compute infrastructure. Across all scenarios, energy-per-operation declines substantially over time; however, aggregate AI electricity demand and emissions diverge sharply depending on model scaling choices and electricity mix assumptions.

The analysis yields several policy-relevant insights. First, efficiency gains alone consistently underperform structural interventions: scenarios relying primarily on hardware and algorithmic improvements reduce emissions by less than 20% relative to baseline outcomes, whereas scenarios combining model consolidation with low-carbon electricity transitions achieve reductions of up to 40%. Second, deployment structure matters quantitatively: pathways emphasizing fewer, larger models result in materially lower operational and embodied emissions than decentralized deployment of many smaller models, holding demand growth constant. Third, electricity system decarbonization emerges as a necessary condition for stabilizing AI-related emissions, particularly in regions hosting energy-intensive training infrastructure. These results suggest that policies prioritizing renewable-powered compute, infrastructure siting aligned with low-carbon grids, and transparency in AI energy use can yield larger emissions reductions than efficiency-oriented measures alone.

All projections remain contingent on modeling assumptions and available data. Efficiency trajectories extrapolate from recent trends and may encounter diminishing returns due to physical, material, or algorithmic constraints, while adoption pathways do not account for abrupt regime shifts driven by policy, geopolitics, or disruptive innovation. Uncertainty also persists in upstream supply chains, particularly regarding hardware lifetimes, utilization rates, and end-of-life management, which are parameterized using generalized estimates. Future work should integrate stochastic adoption dynamics, explicit modeling of circular economy pathways, and adaptive policy scenarios to better capture nonlinear transitions and distributional impacts.

Taken together, the findings do not constitute a prediction of AI's future energy footprint, but rather delineate the range of outcomes implied by plausible deployment and energy system pathways. The results indicate that avoiding high-emissions trajectories will require coordinated action that aligns AI development with electricity system decarbonization and infrastructure planning, rather than reliance on efficiency gains alone.

## Supporting information

**S1 File. Supporting information** file containing detailed methods, extended results, scenario definitions, data sources, and additional tables and figures.
(PDF)

## Author contributions

**Conceptualization:** Murat Kucukvar, Metin Türkay.

**Data curation:** Berke M. Turkay, Ipek Pehlivan.

**Formal analysis:** Berke M. Turkay, Ipek Pehlivan, Nuri C. Onat.

**Investigation:** Berke M. Turkay, Ipek Pehlivan, Nuri C. Onat, Murat Kucukvar, Metin Türkay.

**Methodology:** Berke M. Turkay, Nuri C. Onat, Murat Kucukvar, Metin Türkay.

**Project administration:** Murat Kucukvar, Metin Türkay.

**Resources:** Berke M. Turkay, Nuri C. Onat, Murat Kucukvar, Metin Türkay.

**Software:** Berke M. Turkay, Ipek Pehlivan.

**Supervision:** Nuri C. Onat, Murat Kucukvar, Metin Türkay.

**Validation:** Berke M. Turkay, Ipek Pehlivan, Nuri C. Onat, Murat Kucukvar, Metin Türkay.

**Visualization:** Berke M. Turkay, Ipek Pehlivan.

**Writing – original draft:** Berke M. Turkay, Ipek Pehlivan.

**Writing – review & editing:** Berke M. Turkay, Nuri C. Onat, Murat Kucukvar, Metin Türkay.

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
