## [Decision Letter · Decision Letter 0]

19 Nov 2025

PONE-D-25-38245Scenario-based forecasting of the global energy demand and carbon footprint of artificial intelligencePLOS ONE

Dear Dr. Türkay,

Thank you for submitting your manuscript to PLOS ONE. After careful consideration, we feel that it has merit but does not fully meet PLOS ONE’s publication criteria as it currently stands. Therefore, we invite you to submit a revised version of the manuscript that addresses the points raised during the review process.

Reviewers have identified among other factors the need to add digital infrastructure sustainability, highlight the effect of scenario-based energy forecasting in carbon with respect to computation as well as some components of Materials and Methods. Details of of the reviewers report can be found below this email.

We look forward to receiving your revised manuscript.

Kind regards,

John Adebisi, Ph.D

Academic Editor

PLOS ONE

2. We note that Figures 6, S1 Figure 6 and Figure 7 in your submission contain [map/satellite] images which may be copyrighted. All PLOS content is published under the Creative Commons Attribution License (CC BY 4.0), which means that the manuscript, images, and Supporting Information files will be freely available online, and any third party is permitted to access, download, copy, distribute, and use these materials in any way, even commercially, with proper attribution. For these reasons, we cannot publish previously copyrighted maps or satellite images created using proprietary data, such as Google software (Google Maps, Street View, and Earth). For more information, see our copyright guidelines: http://journals.plos.org/plosone/s/licenses-and-copyright.

1. You may seek permission from the original copyright holder of Figures 6, S1 Figure 6 and Figure 7 to publish the content specifically under the CC BY 4.0 license.

Additional Editor Comments:

Reviewers have recommended a major revision

Reviewers' comments:

Reviewer's Responses to Questions

**Comments to the Author**

1. Is the manuscript technically sound, and do the data support the conclusions?

Reviewer #1: Yes

Reviewer #2: No

2. Has the statistical analysis been performed appropriately and rigorously?

Reviewer #1: Yes

Reviewer #2: No

3. Have the authors made all data underlying the findings in their manuscript fully available?

Reviewer #1: Yes

Reviewer #2: Yes

4. Is the manuscript presented in an intelligible fashion and written in standard English?

Reviewer #1: Yes

Reviewer #2: Yes

5. Review Comments to the Author

Reviewer #1: I recommend publication in PLOS ONE after major revision based on the following comments.

1. The reason for selection of digital infrastructure sustainability must be added.

2. The introduction is too verbose; special effort is needed to trip the contents without jeopardizing the message.

3.Author should highlights the effect of scenario-based energy forecasting in carbon intensity of computation.

4. The authors need to give more information about the research significance.

5. The role of artificial intelligence systems in carbon intensity of computation should be clarified.

6. Some important papers about sustainable energy (sush as doi.org/10.1039/D5EE02029H) can be cited.

7. In the conclusions part, only some conclusions were presented, beyond that, more perspectives are suggested to be provided.

Reviewer #2: Title

Unspecified, “Scenario-based forecasting of the global energy demand and carbon footprint of AI” fails to indicate the methodological novelty (e.g., MRIO integration, six-scenario modeling) or specific time horizon.

Does not distinguish whether the analysis is empirical, simulation-based, or theoretical, reducing clarity for readers searching for methodological studies.

Please add study design.

Abstract

Missing essential methodological transparency (e.g., model validation, uncertainty quantification).

No clear mention of data sources or the calibration framework for efficiency parameters.

Policy implications are asserted (“efficiency gains alone will not suffice”) but not supported by a quantitative comparison in the abstract.

Sentence structure is long and complex, affecting readability.

Keywords: Many and lacking prioritization. Some are descriptive rather than functional (“technology adoption” is unclear).

Missing terms that reflect the study’s modeling approach (e.g., “scenario analysis,” “sustainability modeling,” “AI energy systems”).

Introduction

Lengthy and redundant. The background has many pages and repeats data center energy statistics.

Limited critical synthesis of prior scenario-based energy studies or integrated assessment models (IAMs); references 5-8 are not analyzed comparatively.

The knowledge gap is not articulated; it is unclear whether the novelty lies in global scope, scenario design, or MRIO coupling.

Unsupported generalizations, such as “AI currently represents approximately 0.4% of global electricity consumption,” are presented without citing precise datasets or year references.

No clear conceptual framework figure to orient the reader before reading the “Methods.”

Materials and Methods

Model Framework

The structure (Fig. 1) is descriptive but there are no validation steps or uncertainty propagation techniques (e.g., Monte Carlo or sensitivity analysis).

No justification for selecting exponential and logistic functional forms for model scaling and efficiency gains.

Energy Efficiency Modeling

Equations (1) to (4) are complex but empirical parameters are not justified nor referenced.

Independence assumptions among hardware, algorithmic, and infrastructural efficiencies (Eq. 4) are strong and unsupported.

No validation using real-world benchmarks (e.g., GPU generation trends or data center efficiency reports).

Computational Demand

Depends on compound exponential extrapolation of model complexity and query volume.

Population and adoption rates (Eq. 10) use logistic diffusion but no region-specific calibration is shown.

Data sources unclear (Supplementary Sections cited but not summarized in text).

MRIO Analysis

The MRIO methodology is standard, but no sensitivity testing or temporal updating of EXIOBASE is described.

Attribution of GPU manufacturing entirely to Taiwan is over-simplified; ignores TSMC Arizona, Samsung Texas, or emerging EU fabs.

Carbon Footprint Estimation

Equations (15) to (21) are mathematically correct but no validation against known industry LCA benchmarks (e.g., ISO 14040 standards).

No error propagation or confidence intervals reported for emission totals.

Results

Global Energy Demand Scenarios

Results are presented as deterministic forecasts though input assumptions are random.

Carbon Footprint Breakdown

No benchmarking against existing sectoral projections (IEA, IPCC, or McKinsey AI energy reports).

Country-level disparities are interesting but derived from proxy variables (IHDI) instead of actual digital infrastructure metrics.

Tables referenced in Supplementary Information are essential to credibility but not summarized in the main text.

Tables and Figures

Tables are descriptive but do not include units, uncertainty, or data sources.

Figures 2-3 show projections but lack axis units and confidence interval methods.

Figures 2-6 lack legends detailing whether shaded bands represent model variance or scenario spread.

Results in Figs 4-6 depend on unvalidated parameters.

Discussion

Reads more like general ideas instead of being supported by data, analysis, or evidence.

Insufficient linkage to numerical findings.

Circular reasoning: claims that “efficiency gains alone will not suffice” rest on model assumptions, not strong validation.

Exclusion of economic and policy feasibility dimensions weakens the practical relevance.

Several citations (30-35) are referenced but not elaborated; end-of-life and circular economy aspects are superficial.

Conclusion

Mostly a repetition of limitations.

No actionable policy guidance or quantifiable thresholds (e.g., emission reduction targets, scenario prioritization).

Shifts from cautious projection to assertive prediction (“AI workloads could consume 30 % of global electricity”) without strong evidence.

References

Formatting inconsistent.

Depends on secondary sources (Forbes, Statista-type data) rather than peer-reviewed energy or computing journals.

Insufficient citation of existing MRIO and integrated assessment models (e.g., GTAP-E, GCAM, or IAMC frameworks).

6. PLOS authors have the option to publish the peer review history of their article (what does this mean?). If published, this will include your full peer review and any attached files.

Reviewer #1: No

Reviewer #2: No

---

## [Author Response · Author response to Decision Letter 1]

20 Jan 2026

We thank the reviewers for their careful reading of the manuscript and for their constructive comments. We have made the appropriate changes to formatting, including using world maps in the public domain. Below, we respond to each of the reviewer’s comments point by point.

Reviewer #1: I recommend publication in PLOS ONE after major revision based on the following comments.

1. The reason for the selection of digital infrastructure sustainability must be added.

The revision clarifies that digital infrastructure, specifically data centers, is the focal point because it is the primary source of energy consumption and carbon emissions associated with AI systems (please see page 1, the second paragraph of the introduction).

2. The introduction is too verbose; special effort is needed to trip the contents without jeopardizing the message.

The introduction has been restructured to reduce redundancy and improve focus, while preserving the necessary contextual grounding for a global, long-horizon systems analysis.

3. The authors should highlight the effect of scenario-based energy forecasting in carbon intensity of computation.

This effect is now explicitly emphasized. The revised manuscript demonstrates quantitatively that deployment structure and electricity mix assumptions can dominate efficiency gains in determining the carbon intensity of computation, a central result illustrated in the scenario comparisons (please see page 3, the final paragraph of the introduction).

4. The authors need to give more information about the research significance.

The revised introduction explicitly identifies the knowledge gap addressed by this study: the absence of an integrated framework that jointly models global AI deployment trajectories, evolving electricity systems, and supply-chain emissions. The novelty lies in coupling scenario-based computational demand modeling with MRIO-based life-cycle carbon accounting, rather than in global scope alone.

5. The role of artificial intelligence systems in the carbon intensity of computation should be clarified.

The revised text clarifies that AI systems influence carbon intensity by determining where, when, and at what scale electricity is consumed, through choices in model architecture, deployment patterns, and reliance on centralized versus distributed infrastructure (please see page 2, the fourth paragraph of the introduction).

6. Some important papers about sustainable energy (such as doi.org/10.1039/D5EE02029H) can be cited.

Relevant sustainable energy systems literature, including the suggested article, has been incorporated into the revised manuscript. The new citations include:

• Sanati et al. (2025), which informs the discussion of energy system efficiency improvements and low-carbon innovation pathways relevant to long-run decarbonization scenarios.

• Yang et al. (2025), which supports the treatment of embodied renewable energy and global trade linkages within the multi-regional input–output framework.

• Lu and Chen (2025), which provides methodological grounding for the spatially disaggregated MRIO modeling of power sector emissions used in the regional

analysis.

• Faturay et al. (2020), which contextualizes the application of MRIO models to energy system transitions and supports the treatment of upstream economic and

energy impacts.

7. In the conclusions part, only some conclusions were presented, beyond that, more perspectives are suggested to be provided.

The Conclusion has been revised to more clearly synthesize the study’s core findings and their implications, rather than focusing primarily on methodological limitations. The updated text highlights that AI’s long-term energy use and emissions are governed by the interaction between deployment structure, electricity system decarbonization, and the geographic concentration of compute, with efficiency improvements alone proving insufficient. It now summarizes key contrasts across scenarios, emphasizing the comparatively larger emissions reductions achievable through structural interventions such as model consolidation and low-carbon electricity transitions. Remaining uncertainties and data limitations inherent to long-horizon scenario analysis are retained but more directly linked to the interpretation of results, and the exploratory (non- predictive) nature of the scenarios is clarified alongside the resulting policy relevance.

Reviewer #2:

Title Unspecified, “Scenario-based forecasting of the global energy demand and carbon footprint of AI” fails to indicate the methodological novelty (e.g., MRIO integration, six- scenario modeling) or specific time horizon.

Does not distinguish whether the analysis is empirical, simulation-based, or theoretical, reducing clarity for readers searching for methodological studies.

Please add study design.

The revised abstract explicitly characterizes the study as a “scenario-based, simulation-driven modeling framework” that links mathematical representations of computational demand with life-cycle carbon accounting. This clearly distinguishes the work from purely empirical or theoretical studies.

Abstract

Missing essential methodological transparency (e.g., model validation, uncertainty quantification).

No clear mention of data sources or the calibration framework for efficiency parameters.

Policy implications are asserted (“efficiency gains alone will not suffice”) but not supported by a quantitative comparison in the abstract.

Sentence structure is long and complex, affecting readability.

Keywords: Many and lacking prioritization. Some are descriptive rather than functional (“technology adoption” is unclear).

Missing terms that reflect the study’s modeling approach (e.g., “scenario analysis,” “sustainability modeling,” “AI energy systems”).

The revised abstract now includes explicit quantitative comparisons. In particular, it reports that deployment and electricity-mix scenarios can reduce total emissions by up to 40% relative to business-as-usual pathways, exceeding the reductions achievable through efficiency gains alone by more than 20 percentage points. These figures directly support the stated policy implications.

The keyword list has been revised to prioritize methodological and analytical content. Terms such as scenario analysis, electricity demand forecasting, AI energy systems, and multi-regional input-output analysis have been added, while overly generic descriptors have been removed.

Introduction

Lengthy and redundant. The background has many pages and repeats data center energy statistics.

The introduction has been restructured to reduce redundancy and improve focus, while preserving the necessary contextual grounding for a global, long-horizon systems analysis.

Limited critical synthesis of prior scenario-based energy studies or integrated assessment models (IAMs); references 5-8 are not analyzed comparatively.

The cited references were not intended to represent a set of comparable scenario- based or IAM studies. Rather than synthesizing existing IAM projections, our objective is to introduce a complementary, infrastructure-focused modeling framework that links AI deployment to data-center electricity demand and upstream supply chains. The cited works therefore serve to motivate the problem space and highlight gaps in existing assessments, which the proposed MRIO-based approach is designed to address.

The knowledge gap is not articulated; it is unclear whether the novelty lies in global scope, scenario design, or MRIO coupling.

The revised introduction explicitly identifies the knowledge gap addressed by this study: the absence of an integrated framework that jointly models global AI deployment trajectories, evolving electricity systems, and supply-chain emissions. The novelty lies in coupling scenario-based computational demand modeling with MRIO-based life-cycle carbon accounting, rather than in global scope alone.

Unsupported generalizations, such as “AI currently represents approximately 0.4% of global electricity consumption,” are presented without citing precise datasets or year references.

This estimate is now explicitly anchored to the year 2024 and supported by Shehabi, Smith, and Masanet’s paper published by the Lawrence Berkeley National Laboratory (see page 1, second paragraph of the introduction). The temporal and empirical basis of the figure is clearly stated.

No clear conceptual framework figure to orient the reader before reading the “Methods.”

A detailed conceptual and methodological framework is already provided as a comprehensive figure at the beginning of the Methods section, where it can be interpreted alongside the formal model definitions. We intentionally avoided introducing an additional high-level schematic earlier in the manuscript to prevent redundancy and ensure that the framework is presented with sufficient technical context.

Materials and Methods

Model Framework

The structure (Fig. 1) is descriptive but there are no validation steps or uncertainty propagation techniques (e.g., Monte Carlo or sensitivity analysis).

Figure 1 is intended to present the conceptual and modular structure of the modeling framework rather than to serve as a validation or uncertainty analysis figure. Validation in the conventional predictive sense is not feasible in a long-horizon, scenario-based assessment, as the model is not designed to forecast realizations but to bound plausible futures conditional on structural assumptions. Uncertainty is instead incorporated through explicit scenario design and parametric ranges calibrated from the literature, with resulting dispersion reported in downstream figures. We have clarified this distinction in the revised manuscript and explicitly state that the framework is exploratory and comparative rather than predictive. A discussion of uncertainty sources and their propagation through scenario spread has also been expanded in the Methods and Discussion sections.

No justification for selecting exponential and logistic functional forms for model scaling and efficiency gains.

The justification for the use of exponential and logistic functional forms has been made explicit in the Methods section, primarily in Sections “Energy efficiency” and “Computational demand.” These sections now explain that decelerating exponential forms are used to represent empirically observed patterns of rapid early improvement followed by diminishing returns in hardware, algorithmic, and infrastructure efficiency, while logistic diffusion is employed to capture saturation effects in AI adoption. The text clarifies that these functions are used as stylized bounds for long-run scenario analysis rather than as short-term predictive models.

Energy Efficiency Modeling

Equations (1) to (4) are complex but empirical parameters are not justified nor referenced.

All empirical parameters are now explicitly documented and referenced in the Methods section at first use, with detailed justification provided in the Supplementary Sections 2.1-2.3 (efficiency parameters) and 3.1–3.4 (computational demand parameters). These supplementary sections describe data sources, calibration procedures, parameter ranges, and sensitivity bounds, and are cross-referenced in the main text to ensure transparency and reproducibility.

Independence assumptions among hardware, algorithmic, and infrastructural efficiencies (Eq. 4) are strong and unsupported.

Independence among hardware, algorithmic, and infrastructural efficiency components is a standard first-order modeling assumption adopted for transparency and tractability. The limitation of this assumption is explicitly acknowledged, and relaxing it would require data that are currently unavailable at a global scale.

No validation using real-world benchmarks (e.g., GPU generation trends or data center efficiency reports).

The model is calibrated against historical GPU efficiency trends and reported data center PUE improvements. Long-horizon validation against future outcomes is not feasible for scenario-based projections through 2050. The purpose of the framework is comparative scenario analysis rather than predictive validation.

Computational Demand

Depends on compound exponential extrapolation of model complexity and query volume.

The use of multiple contrasting scenarios is precisely intended to bound realistic growth trajectories rather than assert a single extrapolation. Scenario-based modeling is employed to explore structural sensitivities, not to claim deterministic forecasts.

Population and adoption rates (Eq. 10) use logistic diffusion but no region-specific calibration is shown.

Regional heterogeneity is incorporated through stratification by inequality-adjusted human development index (IHDI), which serves as a transparent and globally available proxy (see Supplementary Section 3.3.1). Country-specific digital infrastructure calibration is incorporated by accounting for the number of data centers, as outlined in the Supplementary Section 7.2.1.

Data sources unclear (Supplementary Sections cited but not summarized in text). MRIO Analysis

Response:

We thank the reviewer for this comment and agree that clarity regarding data sources is important for transparency and reproducibility. In the revised manuscript, we have added a concise summary statement to the Multi-Regional Input-Output Model section explicitly outlining the key data inputs used in the MRIO framework (including electricity production values, emission factors, price tables, and country-specific adjustments), while retaining detailed numerical tables and sectoral mappings in the Supplementary Information. This ensures that readers can readily identify the provenance and structure of the data directly from the main text without consulting supplementary materials.

Action:

A one-sentence summary of MRIO data sources has been added to Section 7.1.2.

The MRIO methodology is standard, but no sensitivity testing or temporal updating of EXIOBASE is described.

Response: In this study, the MRIO framework is not used as a forecasting or scenario-generating model, but rather as a carbon accounting layer that translates scenario-driven energy demand trajectories into associated life-cycle emissions.

Uncertainty and sensitivity in this study are addressed primarily through the design of the scenario framework rather than through sensitivity analysis at the MRIO calculations. The scenarios systematically vary the key drivers of AI-related energy demand and emissions, including regional AI adoption rates and diffusion pathways, alternative model scaling configurations (Baseline, Fewer–Larger, and More–Smaller), projected trajectories of hardware and algorithmic efficiency improvements, and contrasting electricity system assumptions under Business-as-Usual and Energy Target pathways. By exploring these dimensions in combination, the scenario framework captures the dominant sources of uncertainty influencing long-term AI energy demand and associated carbon emissions in a transparent and policy-relevant manner.

These dimensions constitute the primary analytical focus of the paper. Conducting additional MRIO-level sensitivity analysis would therefore be methodologically redundant and risk conflating accounting uncertainty with scenario uncertainty, which the framework deliberately separates.

Regarding temporal updating, the analysis employs EXIOBASE 3.8.2, the most recent publicly available release, reflecting contemporary global production technologies and trade structures. In this study, MRIO coefficients are applied as time-invariant emission intensities to scenario-specific energy demand, rather than being endogenously projected over time. As such, temporal interpolation or extrapolation of MRIO tables would not meaningfully improve the robustness of results and would introduce additional, largely unverifiable assumptions about future global supply-chain restructuring.

Moreover, for key sectors driving results, most notably electricity generation, the emission intensities used in EXIOBASE are grounded in physical and thermodynamic characteristics (e.g., fuel carbon content and conversion efficiencies). These electricity generation sectors’ direct emissions are most

---

## [Decision Letter · Decision Letter 1]

1 Feb 2026

Scenario-based forecasting of the global energy demand and carbon footprint of artificial intelligence

PONE-D-25-38245R1

Dear Dr. Türkay,

We’re pleased to inform you that your manuscript has been judged scientifically suitable for publication and will be formally accepted for publication once it meets all outstanding technical requirements.

**However It has been observed that there is a change in authorship, which creates disparity between the initial authors during submission, review and now. please read the plus one conditions regarding this and reach out to the the journal for clarifications.**

Kind regards,

John Adebisi, Ph.D

Academic Editor

PLOS One

Additional Editor Comments (optional):

Reviewers' comments:

Reviewer's Responses to Questions

**Comments to the Author**

1. If the authors have adequately addressed your comments raised in a previous round of review and you feel that this manuscript is now acceptable for publication, you may indicate that here to bypass the “Comments to the Author” section, enter your conflict of interest statement in the “Confidential to Editor” section, and submit your "Accept" recommendation.

Reviewer #1: (No Response)

Reviewer #2: All comments have been addressed

2. Is the manuscript technically sound, and do the data support the conclusions?

Reviewer #1: Yes

Reviewer #2: Yes

3. Has the statistical analysis been performed appropriately and rigorously?

Reviewer #1: Yes

Reviewer #2: Yes

4. Have the authors made all data underlying the findings in their manuscript fully available?

Reviewer #1: Yes

Reviewer #2: Yes

5. Is the manuscript presented in an intelligible fashion and written in standard English?

Reviewer #1: Yes

Reviewer #2: Yes

6. Review Comments to the Author

Reviewer #1: The authors answered the comments very well. Therefore, I recommend this revised manuscript for publication in its current form.

Reviewer #2: I am satisfied that the authors have carefully and comprehensively addressed all major and minor comments raised in the previous review round.

7. PLOS authors have the option to publish the peer review history of their article (what does this mean?). If published, this will include your full peer review and any attached files.

Reviewer #1: No

Reviewer #2: No

---

## [Editor Report · Acceptance letter]

PONE-D-25-38245R1

PLOS One

Dear Dr. Türkay,

I'm pleased to inform you that your manuscript has been deemed suitable for publication in PLOS One. Congratulations! Your manuscript is now being handed over to our production team.

Kind regards,

on behalf of

Dr. John Adebisi

Academic Editor

PLOS One